# Clutter Cancellation Methods for Small Target Detection Using High-Resolution W-band Radar

**DOI:** 10.3390/s23177557

**Published:** 2023-08-31

**Authors:** Woosung Hwang, Hongje Jang, Myungryul Choi

**Affiliations:** 1Department of EECI Engineering, Hanyang University, Seoul 04763, Republic of Korea; jokersir@gmail.com; 2Division of Electrical Engineering, Hanyang University, Seoul 04763, Republic of Korea; jahoje@hanyang.ac.kr

**Keywords:** radar detection, radar clutter, radar signal processing, least mean squares (LMS), recursive least squares (RLS)

## Abstract

Drones are currently being used for various applications. However, the detection of drones for defense or security purposes has become problematic because of the use of plastic materials and the small size of these drones. Any drone can be placed under surveillance to accurately determine its position by collecting high-resolution data using various detectors such as the radar system proposed in this paper. The W-band radar has a high carrier frequency, which makes it easy to design a wide bandwidth system, and the wideband FMCW signal is suitable for creating high resolution images from a distance. Unfortunately, the huge amounts of data gathered in this way also contain clutter (such as background data and noise) that is usually generated from unstable radar systems and complex environmental factors, and which frequently gives rise to distorted data. Accurate extraction of the position of the target from this big data requires the clutter to be suppressed and canceled, but conventional clutter cancellation methods are not suitable. Four clutter cancellation algorithms are assessed and compared: standard deviation, adaptive least mean squares (LMS), recursive least squares (RLS), and the proposed LMS. The proposed LMS has combined LMS with the standard deviation method. First, the big data pertaining to the target position is collected using the W-band radar system. Subsequently, the target position is calculated by applying these algorithms. The performance of the proposed algorithms is measured and compared to that of the other three algorithms by conducting outdoor experiments.

## 1. Introduction

Wide bandwidth radar technology is becoming essential for the detection of small drones because of the high-resolution data generated by this technology. Linear-frequency-modulated (LFM) chirp technology has been widely used as a reference to detect the target, owing to the many advantages of this technology such as its high resolution and wide bandwidth. In particular, high-resolution data can be easily realized with wide bandwidth technology by using LFM chirp, which de-chirps the received signal by the pulse compression method [1,2,3]. That is, the final output signals contain particular and accurate distance information.

The W-band frequency range (75 to 110 GHz) is suitable for wideband signal processing. When designing a low pass filter (LPF) with the same bandwidth, designing a 100 MHz bandwidth filter for 10 GHz is relatively easy compared with designing a 100 MHz bandwidth filter at a center frequency of 1 GHz. A wide bandwidth and high-resolution radar system are highly important to accurately determine the position of a small target. And an antenna at 94 GHz has a directional beam pattern, which is helpful to detect the small target.

In recent years, drones have become very popular for defense purposes and civilian business and have prompted many laboratories to study these aircrafts. However, it is difficult to detect a small target in a security facility. The detection of invading drones by radar mostly requires the exact position of the drones in the form of accurate reference position information, which is the reason why high-resolution data are required. Acquiring high-resolution data is problematic in that huge amounts of information necessarily contain a large amount of clutter (from noise, trees, buildings, cars, and so on). Experiments in different environments have shown that the small target echo signal is usually buried among the clutter signals; that is, the clutter would have to be removed to identify the weak echo signals reflected from the target. The intensity of the clutter signals resembles that of the target signal or is sometimes even stronger. Distinguishing the target signals from the clutter requires radar signal processing with a clutter cancellation algorithm.

We experimented with whether a target could be found at a distance of up to 100 m using a metal bar with a diameter of 5 cm on the ground. In addition, the results of increasing the clutter signal towards the ground were confirmed in 5.3 and 5.4 to simulate environmental changes and low flying.

Many algorithms have been proposed to reduce clutter [4,5,6,7,8,9,10,11,12,13]. Most of these algorithms are based on adaptive least mean squares (LMS) or recursive least squares (RLS) processing since these methods are efficient and basic. And there are other studies that have improved RLS [9,10]. Most algorithms that employ an adaptive filter are based on LMS [11]. The LMS and RLS adaptive filters are not significantly different, but the processing time differs considerably depending on the value of the constant λ [12].

In this paper, experiments were conducted by simulating a 5 cm drone with a metal pole at a relatively long distance of 100 m [14,15,16,17,18,19,20] and in a similar situation of a 25 cm drone at a 200 m distance. We introduce appropriate methods for clutter and conventional clutter cancellation, and simple techniques that use a low pass filter (LPF) and averaging filter. Then, three conventional algorithms and the proposed algorithm are introduced and tested by processing data collected at outdoor experimental sites under two different conditions: a high and low signal-to-clutter ratio (SCR). Finally, the results obtained with the algorithms were analyzed to assess their performance in terms of clutter cancellation and target position detection.

## 2. The Problem of Low SCR in the Case of Strong Clutter

Figure 1 shows two characteristic types of clutter signals from the sky and a building. When radar transmits a signal to the sky, the intensity of the received clutter signal is lower, without any peak signal. When radar transmits a signal to a building (located approximately 58 m away), clutter signals arise in full range. The clutter also includes signals that are back-scattered from grass, the ground, and so on. Clutter signals that arise within a range of 50 m may be stronger than the target signal, which makes it difficult to detect low-intensity target signals. Under low-SCR conditions, it is challenging to locate a small target. Most importantly, detection of a target necessitates specification of the position of the target by radar. In particular, canceling the background signal (clutter) under outdoor conditions requires the use of several techniques.

Conventional clutter cancellation entails the subtraction of the background data from the received data. However, it is difficult to apply conventional clutter cancellation when the amount of sampled data is massive because of internal harmonic signals, aliasing, and signal vibration. The results in Figure 2b show no difference from the original signal in Figure 2a. To solve this problem, LPF and STD processing are used to generate pre-processed data. The LPF is an averaging filter, and its length is (0,0) to (0,0,0,0,0), respectively. This filter removes the effects of signal vibration and improves the STD processing results.

## 3. Analysis and Comparison of Clutter Cancellation Methods

The input data for processing are generated as shown in Figure 3. The signal undergoes little vibration in the frequency domain. This characteristic is used to extract features of the signal by using a low pass filter. Subsequently, two sets of data (the background signal set and that of the combined background and target signals) become the input data. All processes described in this paper include standard deviation (STD) processing, which serves to extract the features and improve the effectiveness of the process.

### 3.1. Standard Deviation (STD)

STD processing is used to process the combined background and target data, and the background data. The signal of the target position is a singularity point. Thus, STD processing is an effective approach to extract the target data. The STD is calculated using Equation (1):(1)$$\sigma_{i}\left(k\right)=\sum^{N}{(x}_{i}(k)-m)$$
where $\sigma_{i}\left(k\right)$ is the standard deviation value, *m* is the mean value, *N* is the number of data points, *k* is the data length, and the subscript *i* is the data index. Using this processing, range information can be calculated for each range point.

Each set of data has improved characteristics as a result of STD processing with LPF support. The target and background data are used to suppress clutter signals. STD operates as a high-power detection method. The received beat frequency vibrates in the frequency domain, which generates a high value of STD at a high-power frequency. Therefore, subtraction of the background from the integration data enables the target information to be obtained.

### 3.2. Recursive Least Squares (RLS)

The RLS algorithm uses an adaptive filter to remove stationary (clutter) signals [13]. When the target signal appears, RLS processing can extract the target signal. The background data are used as reference data to cancel the clutter signal. The adaptive filter is expressed by the following equation:(2)$${w(n)}_{i}={w(n)}_{i-1}+k\left(n\right)\xi^{*}\left(n\right),$$
where $w\left(n\right)$ is the weight vector, $k\left(n\right)$ is the gain vector, $\xi^{*}(n)$ is the error vector, and *i* is the index of the weight vector. The gain vector k(n) is written as:(3)$$k(n)=\frac{\lambda^{-1}\;P(n-1)\;u(n)}{1+\lambda^{-1}\;u(n)\;P(n-1)\;u(n)}$$
(4)$$\xi \left(n\right)=d\left(n\right)-w^{H}\left(n-1\right)u(n)$$
(5)$$\lambda =\frac{2\xi (n)}{{\left\|u(n)\right\|}^{2}}$$
(6)$$P\left(n\right)=\lambda^{-1}\;P\left(n-1\right)-\lambda^{-1}\;k\left(n\right)u^{H}\left(n\right)P\left(n-1\right),$$
where *u*(*n*) is the input data, *P*(*n*) is the Riccati equation for the RLS algorithm, ${\left(\cdot \right)}^{H}$ is the Hermitian matrix, and *d*(*n*) is the reference data. Equations (2)–(6) are calculated in sequence by the RLS algorithm. The clutter with the canceled data is contained in the error vector because the background data were used as reference and the difference is the target information.

### 3.3. Least Mean Squares (LMS)

LMS processing also uses the background data as a reference input to form an adaptive filter for detecting targets. The adaptive filter is updated with the reference input data to find errors. During this procedure, the filter coefficients are updated [4]. The updating aims to minimize the error power. The minimum mean square error minimizes $J=E\left[e^{2}(n)\right]$ by adjusting $w_{0}$, $w_{1}$, $w_{2}$, ⋯, $w_{N}$. To minimize the value of J, the error equation is derived as follows:(7)$$e\left(n\right)=d\left(n\right)-w_{0}u\left(n\right)-w_{1}u\left(n-1\right)-w_{2}u\left(n-2\right)\cdots w_{N}u\left(n-N\right)$$
where *e*(*n*) is the error data, *d*(*n*) is the reference data input, $w_{N}$ is the window coefficient, *N* is the number of filter length, and *u*(*n*) is the target data input. This error substitutes Equation (7) for Equation (8).
(8)$$J=E[{\left(d\left(n\right)-w_{0}u\left(n\right)-w_{1}u\left(n-1\right)-w_{2}u\left(n-2\right)\cdots w_{N}u\left(n-N\right)\right)}^{2}]$$

Equation (8) is calculated by computing the partial derivative for each window:(9)$$\frac{\partial J}{\partial w_{0}}=-2E\left[u\left(n\right)e(n)\right]\;\frac{\partial J}{\partial w_{1}}=-2E\left[u\left(n-1\right)e(n)\right]\;\frac{\partial J}{\partial w_{N}}=-2E\left[u\left(n-N\right)e\left(n\right)\right]$$

Equation (9) is used to derive the differential vector of the error power. Generally, the value of J is minimized using a gradient-descent algorithm. The adaptive filter moves the coefficients on the other side to the gradient vector to find the 0 vector. The filter is initialized with the values it usually uses such as $w_{1}=1,w_{2}\cdots w_{N}=0$. That equation is:(10)$$w_{k}\left(n+1\right)=w_{k}\left(n\right)-{\left.\mu \frac{\delta J}{\delta w_{k}}\right|}_{w_{k}=w_{k}(n)}=w_{k}\left(n\right)-{\left.\mu E\left[u\left(n-k\right)e\left(n\right)\right]\right|}_{w_{k}=w_{k}\left(n\right)}$$

The window coefficient, $w_{N}$ is updated from each target data input with Equation (10). To use that predictive information, the adaptive filter removes the error with d(n) of the reference data. The equation for error extraction is Equation (11) and the final output y(n) is expressed by Equation (12). Here, the canceled data containing the clutter are contained in an error vector because the background data are used as reference.
(11)$$e\left(n\right)=d\left(n\right)-\sum_{k=0}^{M-1}w_{k}\left(n\right)u(n-k)$$
(12)$$y\left(n\right)=\sum_{k=0}^{M-1}w_{k}\left(n\right)u(n-k).$$

### 3.4. Proposed LMS with STD

We propose a clutter cancellation method based on LMS with an additional STD process for low-SCR signals in the W-band and the configuration is shown in Figure 4.

The proposed LMS process is as follows: The combined target and background data are denoted as input $s_{i}\left(t\right)\;$and the input background data are $b_{i}\left(t\right)$, where *i* is the number of received target data values. Signal processing is performed in the frequency domain and at log scale. Each input is changed as
(13)$$s_{i}\left(t\right)\to S_{i}(f)$$
(14)$$b_{i}\left(t\right)\to B_{i}(f)$$

The FFT outputs of the combined background and target data and the background data are $S_{i}\left(f\right)$ and $B_{i}(f)$, respectively. The LPF is written as $W_{LPF}(f_{c})$.
(15)$$S_{target\_i}\left(f\right)=S_{i}(f)\cdot W_{LPF}(f_{c})$$
(16)$$B_{back\_i}\left(f\right)=B_{i}(f)\cdot W_{LPF}(f_{c})$$
where $S_{target\_i}\left(f\right)$ is the combined background and target data and $B_{back\_i}\left(f\right)$ is the background data. The matrix integration operator is expressed as
(17)$$S_{integ\_i}=\left[\begin{array}{c} S_{target\_i}\left(f\right)\\ B_{back\_i}\left(f\right) \end{array}\right]$$

Then, the standard deviation process follows Equations (1):(18)$${STD}_{input\_i}(n)=\sum_{n=1}^{N}{\left(S_{integ\_i}(n)-m_{i}\right)}^{2}$$
(19)$${STD}_{background\_i}(n)=\sum_{n=1}^{N}{\left(B_{back\_i}(n)-m_{i}\right)}^{2}$$
where $m_{i}$ is the mean value of the *i*th signal and ${STD}_{input\_i}(n)$ is the input used for LMS processing and the adaptive filter. ${STD}_{background\_i}(n)$ is used as the reference and for clutter cancellation. The following equations are the adaptive LMS sequence. $w_{k}\left(n\right)$ is the weight function and is initialized by $w_{k}\left(1\right)=1$, $w_{k}\left(n\right)=\left[0\right](n\neq 1)$.
(20)$$E\left(n\right)={STD}_{background_{i}}\left(n\right)-\sum_{k=0}^{M-1}w_{k}\left(n\right){\;STD}_{input\_i}(n-k)$$
(21)$$w_{k+1}\left(n\right)=w_{k}\left(n\right)-\mu E{\;\left.\left[{STD}_{input_{i}}\left(n-k\right)\;e\left(n\right)\right]\right|}_{w_{k}=w_{k}\left(n\right)}$$
(22)$$y\left(n\right)=\sum_{k=0}^{M-1}w_{k+1}\left(n\right){\;STD}_{input\_i}(n-k),$$
where *μ* is a gradient constant less than 1, E[**∙**] is the mean function, *e*(*n*) is the error, and *y*(*n*) is the output. ${STD}_{background\_i}\left(n\right)$ is used as the reference signal. Therefore, the target signal information is contained in *e*(*n*). The clutter cancellation process to compute the final output is written as follows:(23)$$Clutter\;cancellation=\frac{{STD}_{input\_i}\left(n\right)}{{STD}_{background\_i}\left(n\right)}$$

Clutter cancellation is a dividing operator and is calculated on the dB scale. This operator is combined with the adaptive LMS process:(24)$$X_{filnal\_output}(n)=e\left(n\right)\cdot \frac{{STD}_{input\_i}\left(n\right)}{{STD}_{background\_i}\left(n\right)}$$

Consequently, the proposed LMS is an integration of both the LMS and STD methods. The clutter with the canceled data is contained in the error vector because the background data are used as reference. Therefore, the final output is produced by multiplying the error vector by the output of STD processing. It has a positive effect on both.

## 4. W-band Radar System for Detection of Small Targets

The specifications of the high-resolution radar used to determine the position of the reference target are as follows: This W-band radar has been developed to operate at the center frequency of 94 GHz. Table 1 lists the FMCW (LFM) radar specifications. The output power is 400 mW, 26 dBm. The chirp length is related to the maximum detectable range and range frequency. The chirp length is changeable and, in the case of this radar, 65 μs is selected to detect a target within a range of 100 m. The radar system includes a pair of Cassegrain antennas for transmitting and receiving, respectively [14,15,16].

The output of the IQ modulator is 3.92 GHz (40 MHz bandwidth) and needs an up-convertor to generate the 94 GHz (960 MHz bandwidth) signal. That is, a frequency multiplier of 24 is used to obtain the 94 GHz signal source. Then, this radar system has a 15 cm resolution as follows, where ∆R, c, and BW are resolution, velocity of light, and bandwidth, respectively.
(25)$$∆R=\frac{c}{2BW}$$

The multiplier is designed for microstrip and MEMS and generates the desired W-band signal, which is an FMCW sawtooth signal. Figure 5 shows the block diagram of this radar system. The W-band signal is routed through a waveguide from the multiplier output to the mixer input.

The radar box consists of the two levels shown in Figure 6. The first level contains the FPGA, ADC/DAC converter, and power. And the second level contains the radar systems, which follow the block diagram in Figure 5.

## 5. Experiment Analysis and Discussion

Figure 7 shows the outdoor experimental site. The target is a 5 cm long metal pole for steady RCS size. The distance to the target location will be 50 and 100 m. The playground is surrounded by trees along the edge, which is located more than 120 m away from the radar. Experiments were executed in two ways, i.e., high and low SCR. The attenuator reduces the output power to achieve a low-SCR condition. Additionally, the viewing angle of the radar is low relative to the ground surface [17,18,19,20,21].

### 5.1. High-SCR Case: Target at 50 m

The input data for processing are shown in Figure 8. Near 50 m, the target data appear together with the background data. The noise floor level is −45 dB on average, which is increased by environmental clutter. A strong clutter signal from the ground exists at 60 m.

The detection results of High-SCR and 50 m target are shown in Figure 9. STD processing extracts the target information around 50 m. Although clutter and wide target signals still exist at 50 m, resolutions of these signals are not high. The RLS algorithm produces a higher-quality result than STD. The target signals can be distinguished among the clutter, despite the clutter at 60 m and close to the radar. LMS processing shows low-intensity clutter at 60 m, in which case it has a higher SCR. The proposed LMS has resulted in the highest SCR. The clutter was removed, and the target signal was clearly extracted [22]. However, the target signal was a little wider than the result of LMS. Table 2 shows the SCRs of each processed result, respectively.

When a small target is detected with the above results, the LMS and proposed LMS methods can accurately detect it based on a 50 m distance to the target. However, in the case of STD or RLS, two or more objects are incorrectly recognized as targets, resulting in distance errors in detection.

### 5.2. High-SCR Case: Target at 100 m

The signals of the target at 100 m are detected and are weaker than those of the target at 50 m. In addition, strong clutter exists at 60 m. The input data of this case for processing are shown in Figure 10.

The detection results of High-SCR and 100 m target are shown in Figure 11. The results of STD processing show a higher noise floor than others. The reason is that the target signal is weak while the normalized clutter intensifies. The RLS and LMS algorithms produce poor results, even though the SCR of the RLS result is lower than 0 dB. On the other hand, the proposed LMS obtains the best result in that the SCR is more than 10 dB (Table 3), which is an acceptable result attributable to the co-operation of STD and LMS.

Attempts to detect a small target with the above results would be successful by using the proposed LMS, which could accurately detect it based on the 100 m target. However, LMS and STD incorrectly recognized the target as two targets and RLS detected three targets, resulting in distance errors in detection.

### 5.3. Low-SCR Case: Target at 10 m

This experiment was conducted to assess the performance of the algorithms under low-SCR conditions. The environmental location and setup differed from those of the previous experiment. Therefore, the clutter characteristic is different, with clutter peaks near 10 and 90 m. The input data of this case for processing are shown in Figure 12.

The detection results of Low-SCR and 10 m target are shown in Figure 13. The RLS algorithm failed to detect the target. The STD algorithm identified the target signal as well as several false detection signals. These results suggest that these two algorithms hardly help with detecting targets with the aid of clutter cancellation. In contrast, the LMS and proposed LMS algorithms produced highly intense peak target signals.

Based on the above results for the 10 m target, the proposed LMS can accurately detect a small target. However, STD and RLS failed to detect any targets, and there is a possibility that LMS will mistakenly be understood to recognize two targets (Table 4).

### 5.4. Low-SCR Case: Target at 20 m

Additional experiments with the target at 20 m were conducted in the same location as the 10 m target experiments. Therefore, the environmental conditions giving rise to clutter are very similar. The received data have the same shape as shown in Figure 14.

The detection results of Low-SCR and 20 m are shown in Figure 15. In Table 5, the processing results show that the algorithms delivered similar performances to those in the previous 10 m experiment. The RLS algorithm again failed to detect the target signal. Both the LMS and proposed LMS produced challengeable performance in terms of detecting the target. These algorithms might be suitable for testing the radar resolution.

The proposed algorithm showed a better SCR improvement effect than the conventional algorithm. These results show a sufficient ability to detect small drones. It will be able to effectively detect small drones illegally approaching secure facilities within 100 m. Also, if the RCS of an object is extended from 5 cm to 25 cm at a 200 m distance, it is possible to detect 25 cm sized targets. Additionally, drones can be carrying radar instead of LiDAR. LiDAR can be affected by the environment and is vulnerable to fog or rain due to its low moisture permeability. These influences make it difficult to support the autonomous flight of drones and the measuring of surfaces in adverse conditions. It is thought that this problem can be sufficiently addressed through the W-band radar and algorithm of this study.

## 6. Conclusions

High-resolution radar and clutter cancellation have been studied with the aim of detecting small plastic drones. W-band radar can collect high-resolution data. However, when the SCR is lowered by the clutter signal, the small-target-detection performance is decreased. Therefore, in this paper, the effect of the clutter cancellation algorithm was compared to the effect of SCR. The study compared the ability of the STD, RLS, LMS, and proposed LMS algorithms to detect a target under high- and low-SCR conditions. It was confirmed that the proposed algorithm shows better results in performance improvement compared to the existing algorithms.

Although RLS performed poorly with weak target signals for the 100 m target, the other results were satisfactory. Under low-SCR conditions, the algorithms produced varying results when attempting to suppress clutter. RLS processing failed to suppress clutter and detect the target signal. Although STD processing detected the target signal, significant amounts of clutter prevailed, or false targets were detected. Large amounts of clutter or false detections increase the probability of incorrect detections and poor performance. Lastly, the results obtained with the LMS and proposed LMS had similar characteristics.

Processing with the proposed LMS yielded the best result, which improved the performance of SCR. Thus, this approach to clutter cancellation may be sufficiently accurate to detect a target under low-SCR conditions. Considering that the radar system could measure while rotating, the position of the target could be determined in a 2D space. This could prevent military and security facilities occupying a large space from being invaded by small objects such as drones.

## Figures and Tables

**Figure 1 sensors-23-07557-f001:**
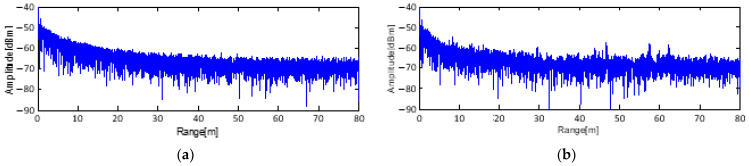
Reflection signals from (**a**) the sky and (**b**) a building.

**Figure 2 sensors-23-07557-f002:**
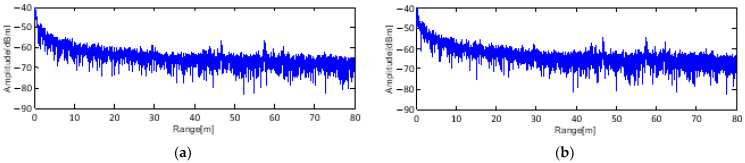
Building reflection signal (**a**) and the difference between the background and target signals (**b**).

**Figure 3 sensors-23-07557-f003:**
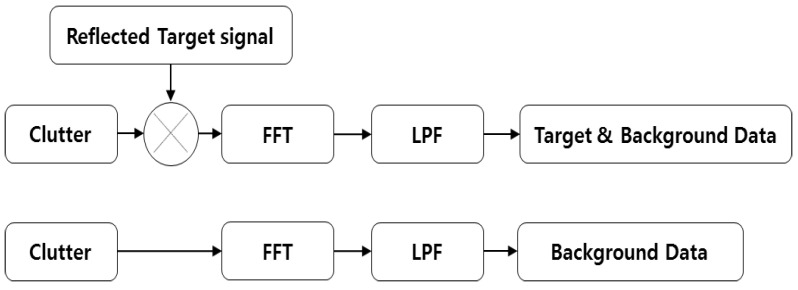
Block diagram of the data acquisition process.

**Figure 4 sensors-23-07557-f004:**
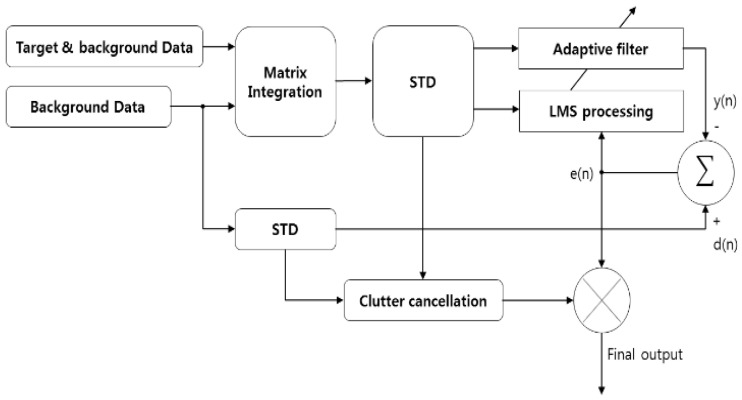
Block diagram of proposed LMS.

**Figure 5 sensors-23-07557-f005:**
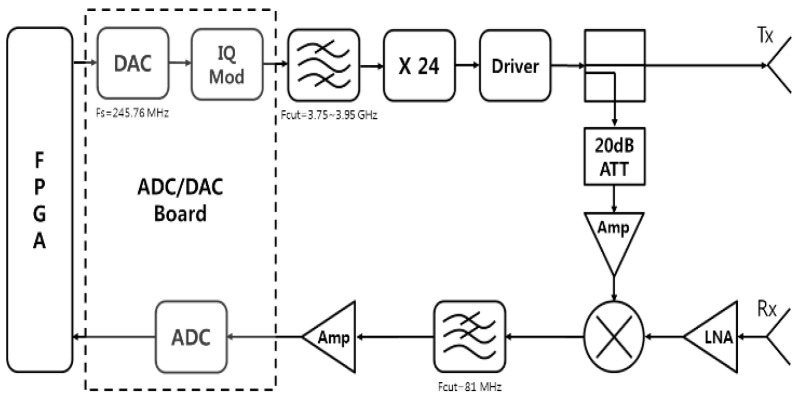
Block diagram of W-band radar system.

**Figure 6 sensors-23-07557-f006:**
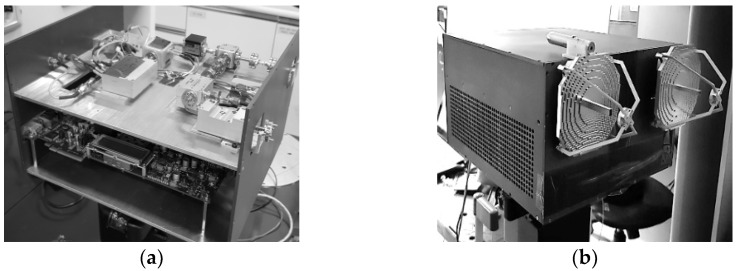
Photographic image of the W-band radar system: (**a**) interior and (**b**) exterior.

**Figure 7 sensors-23-07557-f007:**
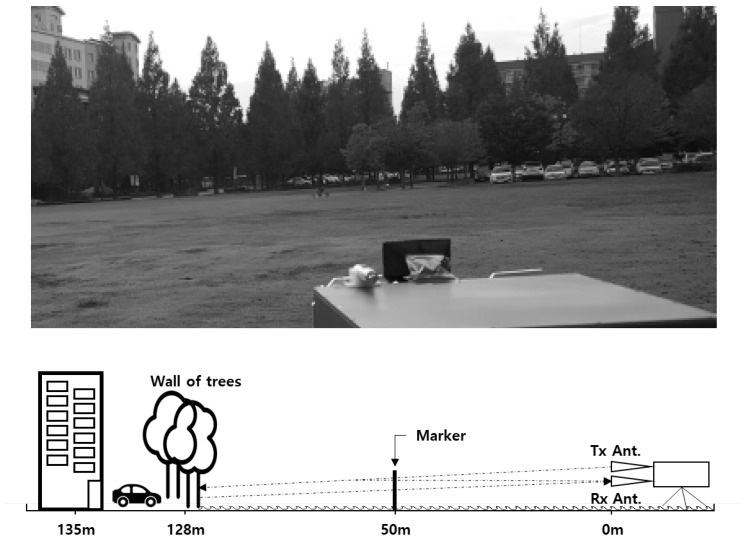
Outdoor experimental site.

**Figure 8 sensors-23-07557-f008:**
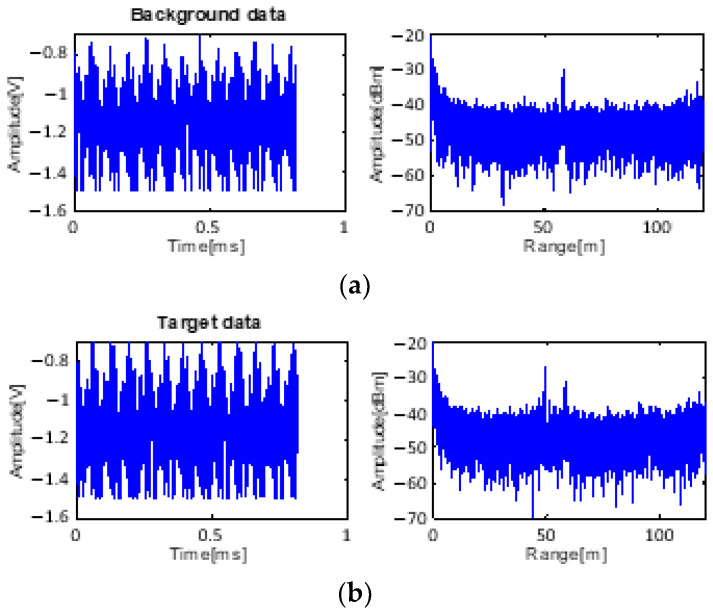
Received data: (**a**) only background data with clutter and (**b**) received data with target at 50 m.

**Figure 9 sensors-23-07557-f009:**
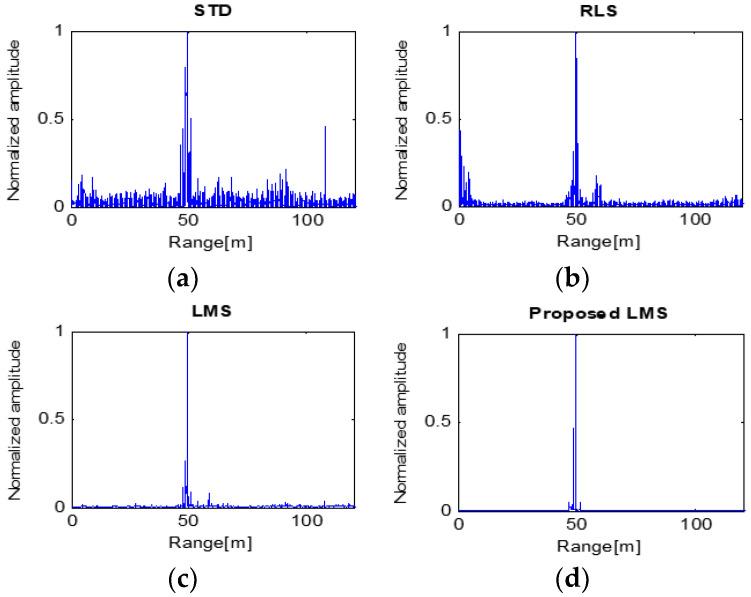
Processing results of four algorithms with the target at 50 m.

**Figure 10 sensors-23-07557-f010:**
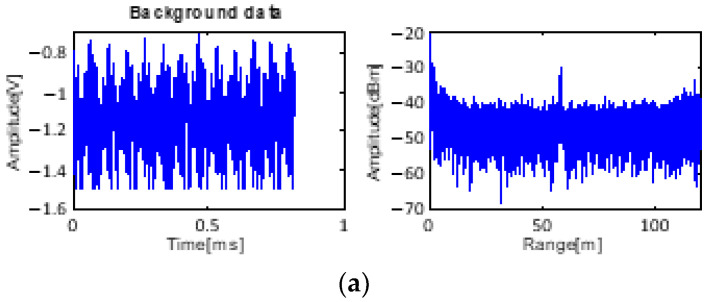
Received data: (**a**) only background data with clutter and (**b**) received data with target at 100 m.

**Figure 11 sensors-23-07557-f011:**
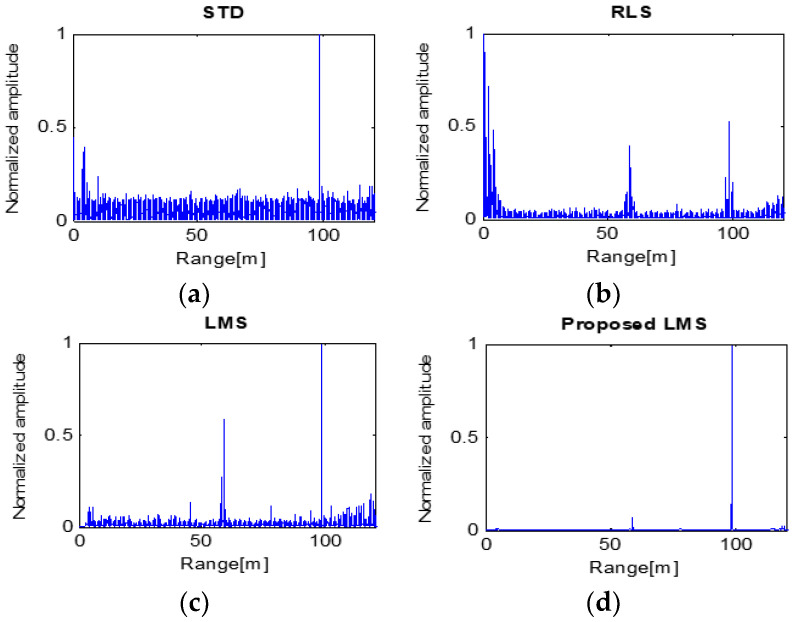
Processing results of four algorithms with the target at 100 m.

**Figure 12 sensors-23-07557-f012:**
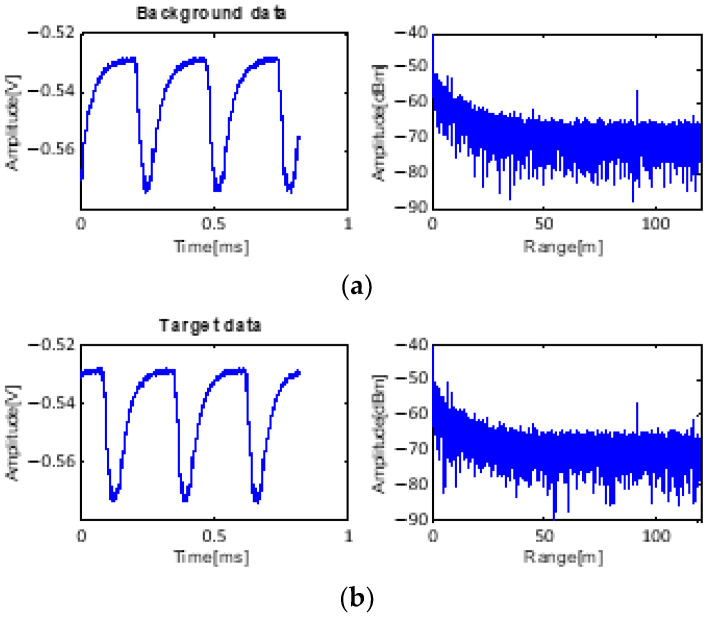
Received data: (**a**) only background data with clutter and (**b**) received data with target at 10 m.

**Figure 13 sensors-23-07557-f013:**
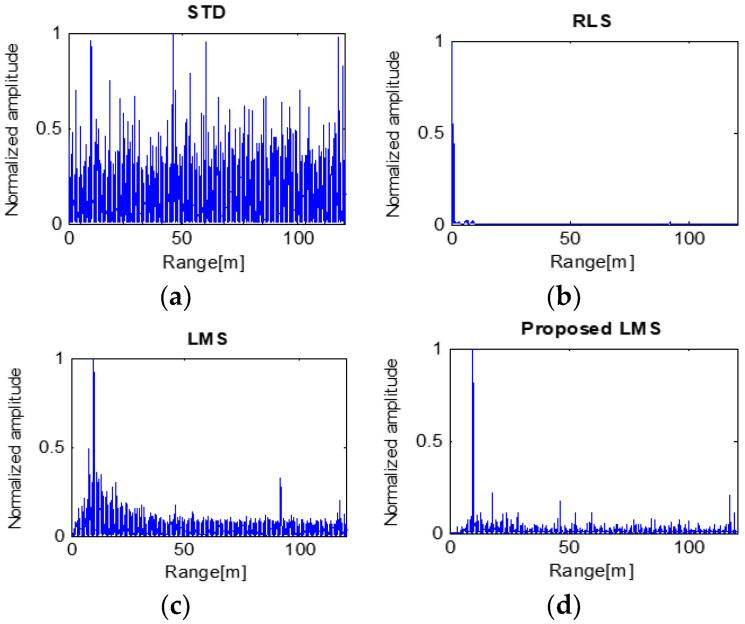
Processing results of four algorithms with the target at 10 m.

**Figure 14 sensors-23-07557-f014:**
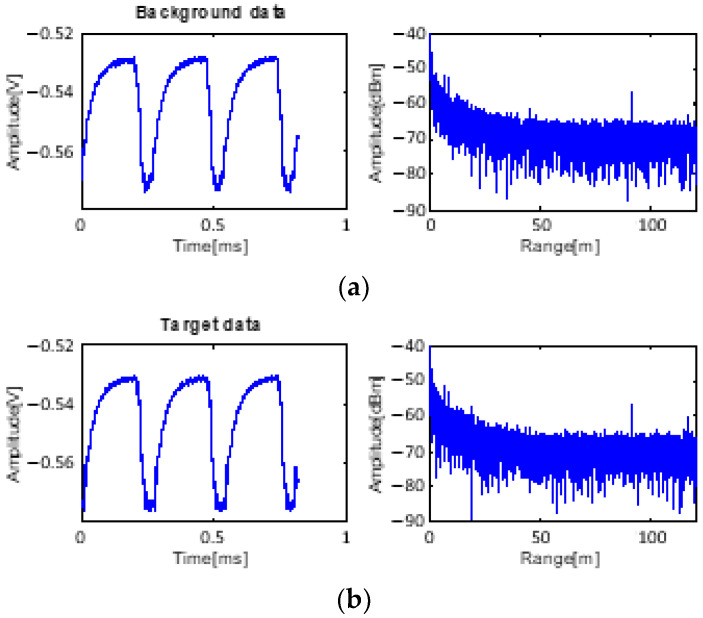
Received data: (**a**) only background data with clutter and (**b**) received data with target at 20 m.

**Figure 15 sensors-23-07557-f015:**
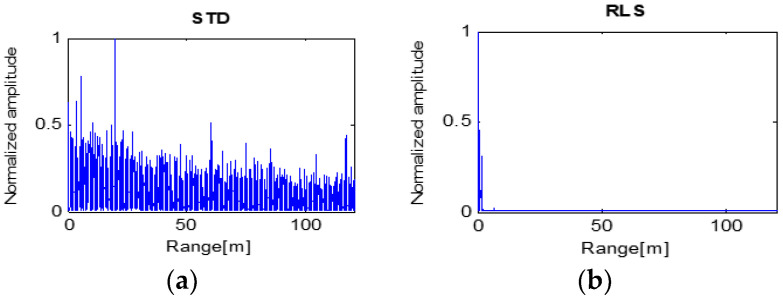
Processing results of four algorithms with the target at 20 m.

**Table 1 sensors-23-07557-t001:** Margin specifications.

Margin	Value
Transmit Frequency	94 GHz
Output Power	400 mW
Waveform	Linear chirp
Chirp Length	65 μs
Bandwidth	960 MHz
Resolution	15 cm
PRF	15 kHz

**Table 2 sensors-23-07557-t002:** Comparison of SCRs with target at 50 m.

Algorithm	Input SCR (dB)	Output SCR (dB)
STD	4.18	3.465
RLS		3.624
LMS		10.612
Proposed LMS		13.452

**Table 3 sensors-23-07557-t003:** Comparison of SCRs with target at 100 m.

Algorithm	Input SCR (dB)	Output SCR (dB)
STD	5.80	4.060
RLS		−2.816
LMS		2.367
Proposed LMS		11.737

**Table 4 sensors-23-07557-t004:** Comparison of SCRs with target at 10 m.

Algorithm	Input SCR (dB)	Output SCR (dB)
STD	−1.31	−0.165
RLS		Miss target
LMS		3.121
Proposed LMS		6.613

**Table 5 sensors-23-07557-t005:** Comparison of SCRs with target at 20 m.

Algorithm	Input SCR (dB)	Output SCR (dB)
STD	−0.94	1.071
RLS		Miss target
LMS		3.035
Proposed LMS		6.203

## Data Availability

Not applicable.

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
