# Peer review of "Clutter Cancellation Methods for Small Target Detection Using High-Resolution W-band Radar"

_sensors, 2023, doi:10.3390/s23177557_

Round 1
Reviewer 1 Report
Please see the comments in the attachment.

Author Response
Suggestion 1: Line 70, the authors say the clutter is within 50 meters. What is the basis for this statement? Need to be clear.
Ans 1. Sentences are added from line 77
Suggestion 2: What is the reflected signal indicated by the red arrow in Figure 8b? Is that a clutter? If it is a clutter, it can be identified in caption.
Ans 2. Yes, it is clutter. We wrote that in the caption.
Suggestion 3: Figure 10 needs to be integrated into a single diagram.
Ans 3. Figure 10 will be edited so that the page does not turn into a single diagram after the review is completed. I will apply the same to other pictures.
Suggestion 4: In Table 1, Table 2, and Table 3, it is recommended to mark the processing times of different algorithms when comparing them. For the future to be applied to the drones, the time efficiency of the algorithm also needs to be considered.
Ans 4. There is no significant difference in process time for 4 algorithms operations. There may be a difference if the number of data used for processing is large enough, but the time difference is meaningless for the 10001 data processing used in this experiment.
Suggestion 5: The author needs to add a discussion section. The application prospect of the new algorithm proposed by the authors need to discussed. Especially in the field of drones carrying radar
Ans 5. Sentences are added from line 77 It was added as a discussion in section 5. This paper aims at radar to detect drones. So, it was written with both sides in mind.
※ Sentences are added from line 319

Reviewer 2 Report
The manuscript proposed a small target detection system using High-Resolution W-band Radar. The overall struct and method description is clear. Several issues:
1. The introduction lacks motivation, and small target detection is not clearly explained. What kind of size means small? From the drone view, if the flying height is large, the big object will also be small. so please clarify the scenario.
2. You only conducted input/ output signal comparison, it can't directly reflect the detection performance or ability. Detection accuracy is missing here.
3. in the abstract, drone application is mentioned, however, the experiment is only ground level from the side-view. if used on the drone, the top view situation, as the ground cover, hills, or cloudy influence, the detection signal may be heavily affected.
4. The reference number is 12, and not included recent years' references, the newest research is missing. Therefore, it is hard to judge the effectiveness of the proposed method.
ok
Author Response
Suggestion 1. The introduction lacks motivation, and small target detection is not clearly explained. What kind of size means small? From the drone view, if the flying height is large, the big object will also be small. so please clarify the scenario.
Ans 1. We have additionally written a clear scenario for small drones. It is added 1. introduction and 5. ~discussion section.
※ Sentences are added from line 319
Suggestion 2. You only conducted input/ output signal comparison, it can't directly reflect the detection performance or ability. Detection accuracy is missing here.
Ans 2. It is judged appropriate to derive the detection accuracy or accuracy rate with various measurements. However, in this paper, it was confirmed that there was a clear difference in target detection by each algorithm, and the performance of the proposed algorithm aimed to show higher performance than other algorithms.
Suggestion 3. In the abstract, drone application is mentioned, however, the experiment is only ground level from the side-view. if used on the drone, the top view situation, as the ground cover, hills, or cloudy influence, the detection signal may be heavily affected.
Ans 3. We added content in introduction. However, in the case of the sky, the distance was limited to about 120 m, so the effect did not appear.
※ Sentences are added from line 56, 319
Suggestion 4. The reference number is 12, and not included recent years' references, the newest research is missing. Therefore, it is hard to judge the effectiveness of the proposed method.
Ans 4. Additional recent papers have been added as reference papers.

Reviewer 3 Report
This paper utilizes W-band radar system to verify various algorithms for small target detection.
Although the paper is well-organized, and there are some algorithm improvements, some major revisions are required before it can be recommended for publication:
(1) Abstract. Authors should clearly state their innovation and conclusion. From the following content authors actually propose improved algorithm, but this was not stated in the abstract. Moreover, why use W-band radar? And what is the new result and advantage? These are not clear in the abstract.
(2) From the result comparison, there are some performance gains. However, these results can only demonstrate the performance of the proposed algorithm, rather than the w-band system. In other words, the advantage brought by the w-band system should be demonstrated.
(3) Relevance should be increased. There are several recent developments on clutter reduction method. For example, there are several sparse methods and their recursive implementations that perform well than the RLS, including
[1] Stoica P, Babu P, Li J. New method of sparse parameter estimation in separable models and its use for spectral analysis of irregularly sampled data[J]. IEEE Transactions on Signal Processing, 2010, 59(1): 35-47.
[2] Zhang Y, Luo J, Li J, et al. Fast inverse-scattering reconstruction for airborne high-squint radar imagery based on Doppler centroid compensation[J]. IEEE Transactions on Geoscience and Remote Sensing, 2021, 60: 1-17.
[3] Stoica, Petre, Zachariah, et al. Online Hyperparameter-Free Sparse Estimation Method[J].IEEE Transactions on Signal Processing: A publication of the IEEE Signal Processing Society, 2015, 63(13):3348-3359.
Author Response
Answer 1 : Based on the comments, the abstract, introduction, and Section 4 have been added or modified.
※ Sentences are added from line 12, 20, 42, 195
Answer 2 : We changed the title to make clear the purpose. And conclusion was added.
※ Sentences are added from line 331, 335
Answer 3 : We add the reference that papers to show recent researches. The papers you recommended showed improved performance over RLS. Our proposed algorithm also showed improved performance compared to the conventional algorithms.

Round 2
Reviewer 2 Report
My concern is solved, no further issues.
The English quality is ok.
Reviewer 3 Report
Authors answered my questions and the manuscript can be accepted.